# On the Power of SVD in the Stochastic Block Model

**Xinyu Mao**
University of Southern California
`xinyumao@usc.edu`

**Jiapeng Zhang**
University of Southern California
`jiapengz@usc.edu`

## Abstract

A popular heuristic method for improving clustering results is to apply dimensionality reduction before running clustering algorithms. It has been observed that spectral-based dimensionality reduction tools, such as PCA or SVD, improve the performance of clustering algorithms in many applications. This phenomenon indicates that spectral method not only serves as a dimensionality reduction tool, but also contributes to the clustering procedure in some sense. It is an interesting question to understand the behavior of spectral steps in clustering problems.

As an initial step in this direction, this paper studies the power of vanilla-SVD algorithm in the stochastic block model (SBM). We show that, in the symmetric setting, vanilla-SVD algorithm recovers all clusters correctly. This result answers an open question posed by Van Vu (Combinatorics Probability and Computing, 2018) in the symmetric setting.

## 1 Introduction

Clustering is a fundamental task in machine learning, with applications in many fields, such as biology, data mining, and statistical physics. Given a set of objects, the goal is to partition them into *clusters* according to their similarities. Objects and known relations can be represented in various ways. In most cases, objects are represented as vectors in $\mathbb{R}^d$, forming a data set $\mathcal{D} \subset \mathbb{R}^d$; each coordinate is called a feature, whose value is directly derived from raw data.

In many applications, the number of features is very large. It has been observed that the performance of classical clustering algorithms such as K-means may be worse on high-dimensional datasets. Some people call this phenomenon *curse of dimensionality in machine learning* [SEK04]. A popular heuristic method to address this issue is to apply dimensionality reduction before clustering. Among tools for dimensionality reduction, it is noted in practice that spectral methods such as *principal component analysis* (PCA) and *singular value decomposition* (SVD) significantly improve clustering results, e.g., [SEK04, KAH19].

A natural question arises: *why do spectral methods help to cluster high-dimensional datasets?* Some practitioners believe one reason is that the spectral method filters some noise from the high-dimensional data [ARS⁺04, SEK04, ZLWZ09, KAH19]. Simultaneously, many theory works also (partially) support this explanation [AFWZ20, EBW18, LZK22, MZ22a]. With this explanation in mind, people analyzed the behavior of spectral-based algorithms with noise perturbation. Based on these analyses, many algorithms were proposed to recover clusters in probabilistic generative models. Among them, a well-studied model is the *signal-plus-noise model*.

**Signal-Plus-Noise model** In this model, we assume that each observed sample $\widehat{v}_i$ has the form $\widehat{v}_i = v_i + e_i$, where $v_i$ is a ground-truth vector and $e_i$ is a random noise vector. For any two sample vectors $\widehat{v}_i, \widehat{v}_j$, if they are from the same cluster, their corresponding ground-truth vectors are identical, i.e., $v_i = v_j$. Signal-plus-noise model is very general; it has plentiful variants with different types of ground-truth vectors and noise distribution. In this paper, we focus on an important instance known as

37th Conference on Neural Information Processing Systems (NeurIPS 2023).

the *stochastic block model* (SBM). Though SBM is not as broad as general signal-plus-noise model, it usually serves as a *benchmark* for clustering and provides *preliminary intuition* about random graphs.

**Stochastic block model**    The SBM is first introduced by [HLL83] and is widely used as a theoretical benchmark for graph clustering algorithms. In the paper, we focus on the *symmetric version of stochastic block model* (SSBM), described as follows. Given a set of $n$ vertices $V$, we uniformly partition them into $k$ disjoint sets (clusters), denoted by $V_1, \ldots, V_k$. Based on this partition, a random (undirected) graph $\widehat{G} = (V, E)$ is sampled in the following way: for all pairs of vertices $u, v \in V$, an edge $\{u, v\}$ is added independently with probability $p$, if $u, v \in V_\ell$ for some $\ell$; otherwise, an edge $\{u, v\}$ is added independently with probability $q$.

We usually assume that $p > q$. The task is to recover the hidden partition $V_1, \ldots, V_k$ from the random graph $\widehat{G}$. We denote this model as $\mathsf{SSBM}(V, n, k, p, q)$.

**SBM as a signal-plus-noise model**    Though SBM was originally designed for graph clustering, we view it as a special form of vector clustering. Namely, given the adjacency matrix of a graph $\widehat{G} \in \{0, 1\}^{V \times V}$, the columns of $\widehat{G}$ form a set of $n = |V|$ vectors. To see that SBM fits into the signal-plus-noise model, note that in the SBM, the adjacency matrix $\widehat{G} \in \{0, 1\}^{V \times V}$ can be viewed as a fixed matrix $G$ plus a random noise, i.e., $\widehat{G} = G + E$, where $G \stackrel{\text{def}}{=} \mathbf{E}\left[\widehat{G}\right]$ is the mean and $E$ is a zero-mean random matrix. More precisely, in the case of SSBM,

$$G_{uv} = \begin{cases} p & \text{if } u, v \in V_\ell \text{ for some } \ell, \\ q & \text{otherwise,} \end{cases} \quad \text{and} \quad E_{uv} = \begin{cases} 1 - G_{uv} & \text{with probability } G_{uv}, \\ -G_{uv} & \text{with probability } 1 - G_{uv}, \end{cases}$$

where the random variables $\{E_{uv} : u \leq v\}$ are independent and $E_{vu} = E_{uv}$ for all $u, v \in V$. [1]

## 1.1 Motivations: Analyzing Vanilla Spectral Algorithms

Since the seminal work by McSherry [McS01], many spectral-based algorithms have been proposed and studied in the SBM [GM05, Vu18, LR15, EBW18, Col19, AFWZ20, MZ22b] and even more general signal-plus-noise models [AFWZ20, EBW18, CTP19, LZK22, MZ22a]. These algorithms are largely based on the spectral analysis of random matrices. The purpose of designing and analyzing such algorithms is twofold.

**Understanding the limitation of spectral-based algorithms**    SBM is specified by parameters, such as $n, k, p, q$ in the symmetric case. Clustering is usually getting harder for larger $k$ and smaller gap $(p - q)$. Many existing works aim to understand in which *regimes* of these parameters it is possible to recover the hidden partition. In this regard, the state-of-the-art bound is given by Vu [Vu18]. Concretely, [Vu18] proved that, in the symmetric setting, there is an algorithm that recovers all clusters if $n \geq C \cdot k \left(\frac{\sigma\sqrt{k} + \sqrt{\log n}}{p - q}\right)^2$, where $\sigma^2 \stackrel{\text{def}}{=} \max\{p(1 - p), q(1 - q)\}$ and $C$ is a constant.

**Understanding spectral-based algorithms in practice**    Besides analyzing spectral algorithms in theory, the other purpose, which is *the primary purpose of this paper*, is to explain the usefulness of such algorithms in practice. Indeed, as we mentioned before, many spectral-based algorithms, as observed in practice, can filter the noise and address the curse of dimensionality [ARS+04, SEK04, ZLWZ09, KAH19]. Some representative algorithms are PCA and SVD. Furthermore, it has been observed that spectral algorithms used in practice, such as PCA or SVD, are usually *very simple*: they just project data points into some lower-dimension subspace, and no extra steps are conducted.

In stark contrast, most of the aforementioned theoretical algorithms have pre-processing or post-processing steps. For example, the idea in [LR15] is that one first applies SVD, and then runs a variant of K-means to clean up the clustering; the main algorithm in [Vu18] partitions the graph into several parts and uses these parts in different ways. As noted in [Vu18], these extra steps are only for the *purpose of theoretical analysis*: From the perspective of algorithm design, these extra steps appear redundant. Later on, [AFWZ20] coined the phrase *vanilla spectral algorithms* to describe spectral algorithms that do not include any additional steps. Both [Vu18] and [AFWZ20] conjectured

---

[1]Assume we fix an order of the vertices in $V$.

that vanilla spectral algorithms are themselves good clustering algorithms. In practice, this is a widely-used heuristic; however, in theory, the analysis of vanilla spectral algorithms is not satisfactory due to the lack of techniques for analysis. We refer to [MZ22b] for a detailed discussion on barriers of the current analysis.

**Why do we study vanilla algorithms?**   Our main focus is particularly on vanilla spectral algorithms for two reasons:

1. Vanilla spectral algorithms are the most popular in practice—no extra steps are widely used. Plus, their performance seems good enough. The lack of theoretical analysis is mostly due to technical obstacles.

2. A vanilla spectral algorithm is often simple and is not specifically designed for theoretical models such as SBM. In contrast, some complicated algorithms use extra steps which are designed for SBM. These steps made the analysis of SBM go through (as commented by [Vu18]). Meanwhile, these extra steps exploit specific structures and may cause 'overfittings' on SBM, which makes these algorithms less powerful in practice.

The main purpose of this paper is to *theoretically understand* the power of *practically successful* vanilla spectral algorithms. To this end, we study SBM as a preliminary demonstration. We *do not* aim to design algorithms for SBM that outperforms existing algorithms.

## 1.2   Our Results

The contribution of this paper is twofold. On the one hand, we show that vanilla algorithms (alg. 1) is indeed a clustering algorithm in the SSBM for a wide range of parameters, breaking previous barrier on analyzing on only constant number of clusters. On the other hand, we provide a novel analysis on matrix perturbation with random noise. We discuss more details on this part in Section 1.4.

Recall that parameters of SBM is specified by $\mathsf{SSBM}(V, n, k, p, q)$, where $n = |V|$. Let $\sigma^2 = \max\{p(1-p), q(1-q)\}$. Our main result is stated below.

**Theorem 1.1.** *There exists a constant $C > 0$ with the following property. In the model* $\mathsf{SSBM}(V, n, k, p, q)$, *if $\sigma^2 \geq C \log n / n$ and $n \geq C \cdot k \left( \frac{\sqrt{kp} \cdot \log^6 n + \sqrt{\log n}}{p - q} \right)^2$, then alg. 1 recovers all clusters with probability $1 - O(n^{-1})$.*

Here we describe the vanilla-SVD algorithm in more detail. Algorithms in [McS01, Vu18, Col19] share a common idea: they both use SVD-based methods to find a clear-cut vector representation of vertices. That is, every node $v \in V$ is associated with a vector $\rho(v)$, and we say a vector representation $\rho$ is *clear-cut* if the following holds for some threshold $\Delta$: if $u, v \in V_\ell$ for some $\ell$, then $\|\rho(u) - \rho(v)\| \leq \Delta/4$; otherwise, $\|\rho(u) - \rho(v)\| \geq \Delta$.

Once a clear-cut representation is found, the clustering task is easy: If the parameters $n, k, p, q$ are all known, we can calculate $\Delta$ and simply decide whether two vertices are in the same cluster based on their distance; in the case where $\Delta$ is unknown, we need one more step. [2] Following [Vu18], we denote by `ClusterByDistance` an algorithm that recovers the partition from a clear-cut representation. One natural representation is obtained by SVD as follows. Let $\widehat{G} \in \{0, 1\}^{V \times V}$ be the adjacent matrix of the input graph, and let $P_{\widehat{G}_k}$ be the orthogonal projection matrix onto the space spanned by the first $k$ eigenvectors of $\widehat{G}$. Then set $\rho(u) \stackrel{\text{def}}{=} P_{\widehat{G}_k} \widehat{G}_u$, where $\widehat{G}_u$ is the column index by $u \in V$. This yields alg. 1, the vanilla-SVD algorithm.

## 1.3   Comparison with Existing Analysis for Vanilla Spectral Algorithms in the SBM

To the best of our knowledge, there are very few works on the analysis of vanilla spectral algorithms [AFWZ20, EBW18, PPV+19]. All of them only apply to the case of $k = O(1)$. In this work, we obtain the first analysis for general parameters $n, k, p, q$, in the symmetric SBM setting.

---

[2]For example, one possible implementation is as follows: create a minimal spanning tree according to the distances under $\rho$, then remove the heaviest $(k - 1)$ edges, resulting in $k$ connected components, and output these components as clusters.

---

**Algorithm 1:** Vanilla-SVD algorithm for graph clustering

---

**1** Input: adjacent matrix $\widehat{G} \in \{0,1\}^{V \times V}$

**2** Output: a partition of $V$

      1. Compute $\rho(u) \stackrel{\text{def}}{=} P_{\widehat{G}_k} \widehat{G}_u$ for each $u \in V$.

      2. Run `ClusterByDistance` with representation $\rho$.

---

**Davis-Kahan approaches**  To study spectral algorithms in signal-plus-noise models, a key step is to understand how random noise perturbs the eigenvectors of a matrix. A commonly-used technical ingredient is the Davis-Kahan $\sin \Theta$ theorem (or its variant). However, this type of approach faces two challenges in the SBM.

- Davis-Kahan leads to *worst-case perturbation* bounds. For perturbations caused by random noises, such as signal-plus-noise models, Davis-Kahan $\sin \Theta$ theorem is sometimes *suboptimal.*

- These $\sin \Theta$ theorems only lead to bound on 2-norm. However, in the SBM analysis, we may need $(2 \to \infty)$-norm bounds. See [CTP19] for more discussions.

Previous works such as [AFWZ20, EBW18, PPV$^+$19] mainly followed this approach. They proposed some novel ideas to (partially) address these two challenges, but only apply to the case of $k = O(1)$. In contrast, our approach, following the power-iteration-based analysis proposed by [MZ22b], completely avoids Davis-Kahan $\sin \Theta$ theorem and can handle the case of $k = \omega(1)$.

**Comparison with [MZ22b]**  Inspired by power iteration methods, Mukherjee and Zhang [MZ22b] proposed a new approach to analyze the perturbation of random matrices. The idea is to approximate the eigenvectors of a matrix by its power. In fact, this method has been widely used in practice as a fast algorithm to approximate eigenvectors. However, there are two limitations of [MZ22b].

- Their analysis requires a nice structure of the mean matrix, i.e., all large eigenvalues are more or less the same.

- Their algorithm is not vanilla as it has a 'centering step'. Moreover, their algorithm requires the knowledge of parameters $p, q, k$, and particularly, the centering step alone requires the knowledge of $q$. In comparison, we only need to know $k$; further, we can also guess $k$ (by checking the number of large eigenvalues) and then make alg. 1 fully parameter-free.

To overcome these limitations, we introduce a novel 'polynomial approximation + entrywise analysis' method, which makes this analysis more robust and requires less structure. More details will be discussed in Section 1.4.

**Regarding parameters in Theorem 1.1**  The difference between the parameters in our results and those in Vu's paper is that we replaced $\sigma$ by $\sqrt{p} \log^6 n$. If $p < 0.9$, $\sigma$ and $\sqrt{p}$ are equal up to constant, so our bound is essentially the same as [Vu18] except for the $\log^6 n$ factor. We believe the extra $\log^6 n$ factor can be improved by future works. This term stems from the new concentration inequality we used as a black box. A refined analysis of this concentration inequality may remove this factor. Here are two example settings of parameters that satisfy the conditions in Theorem 1.1:

- $q = \Theta(\frac{\log n}{n}), p = \Omega(\frac{k \log^6 n}{\sqrt{n}})$;

- $p - q = \Theta(1)$ and $k = O(\frac{\sqrt{n}}{\log^6 n})$.

## 1.4 Proof Outline and Technical Contributions

Let $s_u$ denote the size of the cluster to which $u$ belongs. Assume for now that all $V_i$'s are of size roughly $n/k$. Indeed, this happens with high probability inasmuch as the partition is uniformly sampled.

Our goal is to show that there exists some threshold $\Delta > 0$ such that for every $u, v \in V$: if $u, v \in V_\ell$ for some $\ell$, then $\left\| P_{\widehat{G}_k} \widehat{G}_u - P_{\widehat{G}_k} \widehat{G}_v \right\| \leq \Delta/4$; otherwise, $\left\| P_{\widehat{G}_k} \widehat{G}_u - P_{\widehat{G}_k} \widehat{G}_v \right\| \geq \Delta$. Write $\varepsilon(u) \stackrel{\text{def}}{=} \left\| P_{\widehat{G}_k} \widehat{G}_u - G_u \right\|$. Then $\left| \left\| P_{\widehat{G}_k} \widehat{G}_u - P_{\widehat{G}_k} \widehat{G}_v \right\| - \| G_v - G_u \| \right| \leq \varepsilon(u) + \varepsilon(v)$. Note that $\| G_v - G_u \| = 0$ if $u, v \in V_\ell$ for some $\ell$, otherwise, $\| G_v - G_u \| = (p - q) \cdot \sqrt{s_u + s_v} > (p - q)\sqrt{n/k}$. Therefore, setting $\underline{\Delta = 0.8(p - q)\sqrt{n/k}}$, it suffices to show that $\underline{\varepsilon(u) \leq 0.1(p - q)\sqrt{n/k}}$ for every $u \in V$.

We decompose $\varepsilon(u)$ into two terms:

$$\varepsilon(u) \leq \left\| P_{\widehat{G}_k}(\widehat{G}_u - G_u) \right\| + \left\| (P_{\widehat{G}_k} - I)G_u \right\| = \underbrace{\left\| P_{\widehat{G}_k} E_u \right\|}_{\text{"noise term"}} + \underbrace{\left\| (P_{\widehat{G}_k} - I)G_u \right\|}_{\text{"deviation term"}}. \tag{1}$$

We shall bound the two terms from above separately. Intuitively, the noise term is small means $P_{\widehat{G}_k}$ reduces the noise, while the deviation term is small means $P_{\widehat{G}_k}$ preserves the data.

**Upper bound of the noise term** It is known that $P_{\widehat{G}_k}$ (resp., $P_{G_k}$) can be write as a polynomial of $\widehat{G}$ (resp., $G$). By Weyl's inequality, the eigenvalues of $\widehat{G}$ are not too far from those of $G$. Therefore, in our case, one can find a simple polynomial $\varphi$ which only depends on $G$, such that $\varphi(\widehat{G})$ (resp., $\varphi(G)$) is a good approximation of $P_{\widehat{G}_k}$ (resp., $P_{G_k}$); this is formalized in Lemma 3.2. Then we have the following decomposition: $\left\| P_{\widehat{G}_k} E_u \right\| \leq 2 \left\| \varphi(\widehat{G}) E_u \right\| \leq 2 \| \varphi(G) E_u \| + 2 \left\| \left( \varphi(\widehat{G}) - \varphi(G) \right) E_u \right\|$, where the first inequality follows from Lemma 3.2, which roughly says $\varphi(\widehat{G})$ is a good approximation of $P_{\widehat{G}_k}$.

1. The first term, $\| \varphi(G) E_u \|$, is small with high probability. To see this, we use Lemma 3.2 again: $\| \varphi(G) E_u \| \leq \frac{3}{2} \| P_{G_k} E_u \|$. According to a known result (c.f. Proposition 2.4), $\| P_{G_k} E_u \|$ is small with high probability, largely because the projection $P_{G_k}$ and the vector $E_u$ are independent.

2. The second term is the tricky part, and we draw on an entrywise analysis. Namely, we study every entry of $(\varphi(\widehat{G}) - \varphi(G)) E_u$, using the new inequality from [MZ22b]. See Lemma 3.3 for more details.

The upper bound for the noise term is encapsulated in Lemma 3.4.

**Upper bound of the deviation term** The following argument is reminiscent of [Vu18]. Say $u \in V_\ell$. Note that $G\chi_\ell = \sqrt{s_u} \cdot G_u$ where $\chi_\ell = \frac{1}{\sqrt{s_u}} \cdot 1_{V_\ell}$ is the normalized characteristic vector of $V_\ell$ (i.e., $1_{V_\ell}(v) = 1 \iff v \in V_\ell$). It follows that

$$\left\| (P_{\widehat{G}_k} - I)G \right\|_2 \leq \left\| (P_{\widehat{G}_k} - I)\widehat{G} \right\|_2 + \left\| (P_{\widehat{G}_k} - I)E \right\|_2 \leq \left\| G - \widehat{G} \right\|_2 + \left\| (P_{\widehat{G}_k} - I)E \right\|_2 \leq 2 \| E \|_2,$$

where the second inequality holds because $P_{\widehat{G}_k} \widehat{G}$ is the best $k$-rank approximation of $\widehat{G}$ and $\text{rank}(G) = k$, and in the third inequality, we use $\left\| (P_{\widehat{G}_k} - I) \right\|_2 \leq 1$, as $P_{\widehat{G}_k}$ is a projection matrix. Therefore,

$$\left\| (P_{\widehat{G}_k} - I)G_u \right\| = \frac{1}{\sqrt{s_u}} \left\| (P_{\widehat{G}_k} - I)G\chi_u \right\| \leq \frac{1}{\sqrt{s_u}} \left\| (P_{\widehat{G}_k} - I)G \right\|_2 \leq \frac{2 \| E \|_2}{\sqrt{s_u}}. \tag{2}$$

A typical result in random matrix theory (c.f. Proposition 2.3) states that with high probability, $\| E \|_2 = O(\sqrt{n})$. Combining Equation (2) and $s_u \approx n/k$, we get $\left\| (P_{\widehat{G}_k} - I)G_u \right\| = O(\sqrt{k})$. And by our assumption on $n$, we have $\sqrt{k} = o((p - q)n/k) = o(\Delta)$.

**Technical contribution** The major novelty of our analysis is using the polynomial $\varphi$. [MZ22b] used a centering step to make the mean matrix nicely structured, while in our analysis, we used polynomial approximation to address this issue. Another difference is that in [MZ22b], the centering

step appears explicitly in the algorithm. By contrast, our polynomial approximation only appears in the analysis — the algorithm is vanilla.

As a byproduct, we developed new techniques for studying *eigenspace perturbation*, a typical topic in random matrix theory. Our high-level idea is "polynomial approximation + entrywise analysis". That is, we reduce the analysis of eigenspace perturbation to the analysis of a simple polynomial (of matrix) under perturbation. We have more tools to deal with the latter.

## 1.5 Discussion and Future Directions

In this paper, we studied the behavior of vanilla-SVD in the SSBM, a benchmark signal-plus-noise model widely studied in random matrix theory. We showed that vanilla-SVD indeed filters noise in the SSBM. In fact, our analysis technique, 'polynomial approximation + entrywise analysis', is not very limited to SSBM. A direct and interesting question yet to be answered is: Can our method be extended to prove that vanilla-SVD works in the general SBM where partitions are not uniformly sampled and edges appear with different probabilities? Moreover, our method may be useful for analyzing some other realistic, probabilistic models such as the factor model — a model which has been widely used in economics and model portfolio theory.

In the long term, it would be very interesting to understand the behavior of vanilla spectral algorithms on real data: 1) Why does it succeed in some applications? 2) How could we fix it if it has failed in other cases? A deeper understanding of vanilla spectral algorithms will provide guidelines for using them in many machine learning tasks.

## 2 Preliminaries

**Notations** Let $\mathbf{1}_n$ denote the $n$-dimensional vector whose entries are all 1's, and let $J_n$ be the $n \times n$ matrix whose entries are all 1's. Let $s_u$ denote the size of the cluster to which $u$ belongs. For a matrix $A$, $A[i]$ denotes the row of $A$ indexed by $i$, and $A_i$ denotes the column indexed by $i$; $\lambda_i(A)$ is the $i$-th largest eigenvalue of $A$; let $P_{A_k}$ denote the orthogonal projection matrix onto the space spanned by the first $k$ eigenvectors of $A$. For a vector $x \in \mathbb{R}^n$, $\|x\| \stackrel{\text{def}}{=} \sqrt{x_1^2 + \cdots + x_n^2}$ denotes the Euclidean norm.

**Definition 2.1** (Matrix operator norms). Let $A \in \mathbb{R}^{n \times n}$. Define $\|A\|_2 \stackrel{\text{def}}{=} \max_{\|x\|=1} \|Ax\|$ and $\|A\|_{2\to\infty} \stackrel{\text{def}}{=} \max_{x:\|x\|=1} \|Ax\|_\infty$.

**Proposition 2.1** (e.g., [CTP19]). *For all matrices $A, B \in \mathbb{R}^{n \times n}$, it holds that (1) $\|A\|_{2\to\infty} = \max_{i \in [n]} \|A[i]\|$; (2) $\|AB\|_{2\to\infty} \le \|A\|_{2\to\infty} \|B\|_2$.*

**Proposition 2.2** (Weyl's inequality). *For all $A, E \in \mathbb{R}^{n \times n}$, we have $|\lambda_i(A) - \lambda_i(A + E)| \le \|E\|_2$.*

**Proposition 2.3** (Norm of a random matrix [Vu18]). *There is a constant $C_0 > 0$. Let $E$ be a symmetric matrix whose upper diagonal entries $e_{ij}$ are independent random variables where $e_{ij} = 1 - p_{ij}$ or $-p_{ij}$ with probabilities $p_{ij}$ and $1 - p_{ij}$ respectively, where $p_{ij} \in [0, 1]$. Let $\sigma^2 := \max_{ij}\{p_{ij}(1 - p_{ij})\}$. If $\sigma^2 \ge C_0 \log n/n$, then $\mathbf{Pr}[\|E\|_2 \ge C_0 \sigma n^{1/2}] \le n^{-3}$.*

**Proposition 2.4** (Projection of a random vector, lemma 2.1 in [Vu18]). *There exists a constant $C_1$ such that the following holds. Let $X = (\xi_1, \ldots, \xi_n)$ be a random vector in $\mathbb{R}^n$ whose coordinates $\xi_i$ are independent random variables with mean 0 and variance at most $\sigma^2 \le 1$. Assume furthermore that the $\xi_i$ are bounded by 1 in absolute value. Let $H$ be a subspace of dimension $d$ and let $\Pi_H \xi$ be the length of the orthogonal projection of $\xi$ onto $H$. Then $\mathbf{Pr}\left[\Pi_H X \ge \sigma\sqrt{d} + C_1\sqrt{\log n}\right] \le n^{-3}$.*

**Proposition 2.5.** *For $a \in [0, 2]$ and $r \in \mathbb{N}$, if $|a - 1| \le \delta < \frac{1}{2r}$, then $|a^r - 1| \le 2r\delta$.*

## 3 Analysis of Vanilla SVD Algorithm

Write $s_i \stackrel{\text{def}}{=} |V_i|$. We say the partition $V_1, \ldots, V_k$ is *balanced* if $\left(1 - \frac{1}{16\log n}\right)\frac{n}{k} \le s_i \le \left(1 + \frac{1}{16\log n}\right)\frac{n}{k}, \forall i \in [k]$. By Chernoff bound, the partition $V_1, \ldots, V_k$ is balanced with probability at least $1 - n^{-1}$; hence, we assume that the partition is balanced in the following argument. Since $\sigma^2 \ge C \log n/n$, the event $\|E\| = O(\sqrt{n})$ holds with high probability (see Proposition 2.3).

Recall the decomposition into deviation term and noise term in Equation (1). We first state our upper bound of the deviation term, which readily follows from the argument in Section 1.4, and the complete proof is in Appendix B.

**Lemma 3.1** (Upper bound of deviation term). *Let $C_0$ be the constant in Proposition 2.3. If the partition is balanced and $n \geq 10^4 \cdot C_0^2 \frac{k^2 \sigma^2}{(p-q)^2}$, then with probability at least $1 - n^{-3}$ we have*
$$\left\| (P_{\widehat{G}_k} - I) G_u \right\| \leq 0.04(p-q)\sqrt{n/k}, \forall u \in V.$$

Section 3.1 and Section 3.2 lead to an upper bound of the noise term, and Section 3.3 is the proof of main theorem.

## 3.1 An Approximation of $P_{G_k}$ and $P_{\widehat{G}_k}$

In order to give some intuition on the choice of $\varphi$, we first analyze the spectrum of $G$, and the result is summed up in Theorem 3.1.

**The eigenvalues of $G$**   Note that $G = H + q\mathbf{1}_n \mathbf{1}_n^\top$, where $H = \begin{pmatrix} (p-q)J_{s_1} & \mathbf{0} & \mathbf{0} & \mathbf{0} \\ \mathbf{0} & (p-q)J_{s_2} & \mathbf{0} & \mathbf{0} \\ & & \ddots & \mathbf{0} \\ \mathbf{0} & \mathbf{0} & \mathbf{0} & (p-q)J_{s_k} \end{pmatrix}$.

Without loss of generality, assume that $s_1 \geq s_2 \geq \cdots \geq s_k$. It is easy to see that the eigenvalues of $H$ are $(p-q)s_1, \ldots, (p-q)s_k, 0$. Viewing $G$ as a rank-one perturbation of $H$, we have the following theorem that characterizes eigenvalues of $G$. Its proof, in Appendix C, readily follows from a theorem in [BNS79], which studies eigenvalues under rank-one perturbation.

**Theorem 3.1.** *Write $s_i \stackrel{\text{def}}{=} |V_i|$ and assume that $s_1 \geq s_2 \geq \cdots \geq s_k$. Define $\delta_i \stackrel{\text{def}}{=} \lambda_i(G) - (p-q)s_i$, then (1) $\delta_i \geq 0$ and $\sum_{i=1}^k \delta_i = nq$; (2) $\lambda_1(G) \geq nq + (p-q)\frac{n}{k}$, and hence $\sum_{i=2}^k \delta_i \leq (p-q)(s_1 - \frac{n}{k})$.*

**The choice of the polynomial $\varphi$**   Let $\mu \stackrel{\text{def}}{=} (p-q)\frac{n}{k}$, and let $\psi(t)$ be the quadratic polynomial such that $\psi(\lambda_1(G)) = \psi(\mu) = 1, \psi(0) = 0$, i.e., $\psi(t) \stackrel{\text{def}}{=} -\frac{1}{\lambda_1(G)\mu}(t - \lambda_1(G))(t - \mu) + 1 \stackrel{\text{def}}{=} At^2 + Bt$, where $A = -\frac{1}{\lambda_1(G)\mu}, B = \frac{1}{\lambda_1(G)} + \frac{1}{\mu}$. Finally, let $\varphi(t) \stackrel{\text{def}}{=} (\psi(t))^r$ where $r \stackrel{\text{def}}{=} \log n$.

Here we give some intuition for the choice of $\varphi$. Let $\widehat{G} = \sum_{i=1}^n \widehat{\lambda}_i v_i v_i^\top$ be the spectral decomposition of $\widehat{G}$. Then $\varphi(\widehat{G}) = \sum_{i=1}^n \varphi(\widehat{\lambda}_i) v_i v_i^\top, P_{\widehat{G}_k} = \sum_{i=1}^k v_i v^\top$. The spectral decomposition of $\varphi(\widehat{G}) - P_{\widehat{G}_k}$ is $\varphi(\widehat{G}) - P_{\widehat{G}_k} = \sum_{i=1}^k (\varphi(\widehat{\lambda}_i) - 1) v_i v^\top + \sum_{i=k+1}^n \varphi(\widehat{\lambda}_i) v_i v^\top$. Hence,

$$\left\| \varphi(\widehat{G}) - P_{\widehat{G}_k} \right\|_2 = \max\{|\varphi(\widehat{\lambda}_1) - 1|, \ldots, |\varphi(\widehat{\lambda}_k) - 1|, |\varphi(\widehat{\lambda}_{k+1})|, \ldots, |\varphi(\widehat{\lambda}_n)|\}. \quad (3)$$

Recall that $\widehat{\lambda}_i - \lambda_i(G)$ is bounded by Weyl's inequality. Plus, when the partition is balanced, Theorem 3.1 shows that the eigenvalues of $G$ is nicely distributed: Except for $\lambda_1(G)$, other eigenvalues are all close to $\mu$. Hence, our choice of $\varphi$ makes $\left\| \varphi(\widehat{G}) - P_{\widehat{G}_k} \right\|_2$ small, and thus $\varphi(\widehat{G})$ is a good approximation of $P_{\widehat{G}_k}$. Formally, we have the following lemma.

**Lemma 3.2** (Polynomial approximation). *Assume that the partition is balanced and $n \geq 10^4 \cdot C_0^2 \cdot \frac{k^2 \cdot p \cdot \log n}{(p-q)^2}$, where $C_0$ is the constant in Proposition 2.3. Then with probability at least $1 - n^{-3}$, it holds that for all $x \in \mathbb{R}^n$, $\frac{1}{2} \left\| P_{\widehat{G}_k} x \right\| \leq \left\| \varphi(\widehat{G}) x \right\| \leq \frac{3}{2} \left\| P_{\widehat{G}_k} x \right\| + \|x\| / n^{\log \log n}$, and $\frac{1}{2} \|P_{G_k} x\| \leq \|\varphi(G)x\| \leq \frac{3}{2} \|P_{G_k} x\|$.*

*Proof.* Let $G = \sum_{i=1}^k \lambda_i u_i u_i^\top$ (resp., $\widehat{G} = \sum_{i=1}^n \widehat{\lambda}_i v_i v_i^\top$) be the spectral decomposition of $G$ (resp., $\widehat{G}$). We shall use the following claim.

**Claim 3.1.** *The following holds with probability $1 - n^{-3}$ (over the choice of $E$): for every $i \in [k]$, $\left| \varphi(\widehat{\lambda}_i) - 1 \right| < \frac{1}{2}, |\varphi(\lambda_i) - 1| < \frac{1}{2}$; and for every $i = k+1, \ldots, n, \left| \varphi(\widehat{\lambda}_i) \right| < n^{-\log \log n}$.*

Fix $x \in \mathbb{R}^n$. On the one hand, $\frac{1}{2} \le \varphi(\widehat{\lambda}_i) \le \frac{3}{2}, \forall i \in [k]$, and hence

$$\left\| \varphi(\widehat{G})x \right\|^2 = \sum_{i=1}^n \varphi(\widehat{\lambda}_i)^2 \langle x, v_i \rangle^2 \ge \sum_{i=1}^k \varphi(\widehat{\lambda}_i)^2 \langle x, v_i \rangle^2 \ge \sum_{i=1}^k \frac{1}{4} \langle x, v_i \rangle^2 = \frac{1}{4} \left\| P_{\widehat{G}_k} x \right\|^2,$$

which means $\left\| \varphi(\widehat{G})x \right\| \ge \frac{1}{2} \left\| P_{\widehat{G}_k} x \right\|$. On the other hand,

$$\left\| \varphi(\widehat{G})x \right\|^2 = \sum_{i=1}^n \varphi(\widehat{\lambda}_i)^2 \langle x, v_i \rangle^2 \le \sum_{i=1}^k \left( \frac{3}{2} \right)^2 \langle x, v_i \rangle^2 + \sum_{i=k+1}^n \frac{\langle x, v_i \rangle^2}{n^{2 \log \log n}} \le \frac{9}{4} \left\| P_{\widehat{G}_k} x \right\|^2 + \frac{\|x\|^2}{n^{2 \log \log n}}.$$

Since $\sqrt{a+b} \le \sqrt{a} + \sqrt{b}$, we have $\left\| \varphi(\widehat{G})x \right\| \le \frac{3}{2} \left\| P_{\widehat{G}_k} x \right\| + \frac{\|x\|}{n^{\log \log n}}$. This establishes the first part.

Note that $\|\varphi(G)x\| = \sqrt{\sum_{i=1}^k \varphi(\lambda_i)^2 \langle x, u_i \rangle^2}$ and we also have $\frac{1}{2} \le \varphi(\lambda_i) \le \frac{3}{2}, \forall i \in [k]$, and thus similar argument goes for $G$. This finishes the proof of Lemma 3.2.

It remains to prove Claim 3.1. The claim readily follows from the choice of $\varphi$ and the fact that $\lambda_i, \widehat{\lambda}_i$ are close. A complete proof can be found in Appendix C. $\qquad \square$

## 3.2 The Upper Bound of the Noise Term

According to Equation (1), in order to derive an upper bound of $\|P_{G_k} E_u\|$, it remains to bound $\left\| \left( \varphi(\widehat{G}) - \varphi(G) \right) E_u \right\|$ from above. This is done by the following lemma.

**Lemma 3.3.** *Let $C_0$ be the constant in Proposition 2.3. Assume that the partition is balanced and $n \ge (100 + C_0)^2 \cdot \frac{k^2 \cdot p \cdot \log^{12} n}{(p-q)^2}$. For every $u \in V$, it holds that*

$$\Pr_E \left[ \left\| \left( \varphi(\widehat{G}) - \varphi(G) \right) E_u \right\| \le C_2(\sqrt{kp} \log^2 n) + \frac{1}{\log n}) \right] \ge 1 - O(n^{-2}),$$

*where $C_2 \stackrel{\text{def}}{=} 7 \cdot 10^6$ is a constant.*

Combining Lemma 3.2 and Proposition 2.4, we get an upper bound of the noise term:

**Lemma 3.4** (Upper bound of noise term)**.** *Let $C_0$ be the constant in Proposition 2.3. Assume that $n \ge (100 + C_0)^2 \cdot \frac{k^2 \cdot p \cdot \log^{12} n}{(p-q)^2}$. Then with probability at least $1 - O(n^{-1})$, we have $\left\| P_{\widehat{G}_k} E_u \right\| \le C_3(\sqrt{kp} \log^2 n + \sqrt{\log n})$ for all $u \in V$, where $C_3$ is a constant.*

The proof of Lemma 3.3 is deferred to Section 4. We use it to prove Lemma 3.4 here.

*Proof of Lemma 3.4.* It follows from Lemma 3.2 that

$$\left\| P_{\widehat{G}_k} E_u \right\| \le 2 \left\| \varphi(\widehat{G}) E_u \right\| \le 2 \left\| \left( \varphi(\widehat{G}) - \varphi(G) \right) E_u \right\| + 2 \|\varphi(G) E_u\|$$
$$\le 2 \left\| \left( \varphi(\widehat{G}) - \varphi(G) \right) E_u \right\| + 3 \|P_{G_k} E_u\|.$$

By Proposition 2.4, with high probability at least $1 - n^{-1}$, $\|P_{G_k} E_u\|$ is bounded by $\sigma \sqrt{k} + C_1 \sqrt{\log n}$, where $C_1$ is a universal constant. Meanwhile, by Lemma 3.3 and union bound over all $u$, with probability at least $1 - O(n^{-1})$, $\left\| \left( \varphi(\widehat{G}) - \varphi(G) \right) E_u \right\| \le 7 \cdot 10^6 (\sqrt{kp} \log^2 n + 1/\log n)$ for every $u \in V$. Therefore, with probability $1 - O(n^{-1})$, it holds that $\left\| P_{\widehat{G}_k} E_u \right\| \le 1.4 \times 10^7 \sqrt{kp} \log^2 n + 3\sigma \sqrt{k} + 3C_1 \sqrt{\log n}$ for all $u \in V$. Setting $C_3 \stackrel{\text{def}}{=} (1.4 \times 10^7 + 3 + 3C_1)$, we have the desired result. $\qquad \square$

### 3.3 Putting It Together

Now we are well-equipped to prove Theorem 1.1.

*Proof of Theorem 1.1.* Let $C \stackrel{\text{def}}{=} (100 + 100C_0 + 100C_3)^2$, where $C_0, C_3$ are the constants in Proposition 2.3 and Lemma 3.4. By our assumption on $n$, we have $(p - q)\sqrt{n/k} > 100C_3(\sqrt{kp}\log^6 n + \sqrt{\log n})$. It is easy to verify $n$ satisfies the conditions in Lemma 3.4 and Lemma 3.1.

Write $\Delta \stackrel{\text{def}}{=} 0.8(p - q)\sqrt{n/k}$. We aim to show that for every $u, v \in V$: if $u, v \in V_\ell$ for some $\ell$, then $\left\| P_{\widehat{G}_k}\widehat{G}_u - P_{\widehat{G}_k}\widehat{G}_v \right\| \leq \Delta/4$; otherwise, $\left\| P_{\widehat{G}_k}\widehat{G}_u - P_{\widehat{G}_k}\widehat{G}_v \right\| \geq \Delta$. Then by calling `ClusterByDistance`, alg. 1 recovers all large clusters correctly.

Let $\varepsilon(u) \stackrel{\text{def}}{=} \left\| P_{\widehat{G}_k}\widehat{G}_u - G_u \right\|$. According to the argument in Section 1.4, it suffices to show that $\varepsilon(u) \leq 0.1(p - q)\sqrt{n/k}$ for all $u \in V$. We further decompose $\varepsilon(u)$ into noise term and deviation term, i.e., $\varepsilon(u) \leq \mathsf{noise}(u) + \mathsf{dev}(u)$, where $\mathsf{noise}(u) \stackrel{\text{def}}{=} \left\| P_{\widehat{G}}E_u \right\|$ and $\mathsf{dev}(u) \stackrel{\text{def}}{=} \left\| (P_{\widehat{G}} - I)G_u \right\|$. By Lemma 3.4 and Lemma 3.1, with probability at least $1 - O(n^{-1})$, the following hold for all $u \in V$: (1) $\mathsf{noise}(u) \leq C_3(\sqrt{kp}\log^2 n + \sqrt{\log n}) \leq 0.01(p - q)\sqrt{n/k}$; (2) $\mathsf{dev}(u) \leq 0.04(p - q)\sqrt{n/k}$. Therefore, with probability at least $1 - O(n^{-1})$, we indeed have $\varepsilon(u) \leq 0.1(p - q)\sqrt{n/k}, \forall u \in V$. This completes the proof. $\qquad\square$

## 4  Proof of Lemma 3.3: Entrywise Analysis

This section is dedicated to proving Lemma 3.3.

Since both $(\varphi(\widehat{G}) - \varphi(G))$ and $E$ are symmetric, we have $\left\| (\varphi(\widehat{G}) - \varphi(G))E_u \right\| \leq \left\| E(\varphi(\widehat{G}) - \varphi(G)) \right\|_{2\to\infty}$. The high-level idea is to write $E(\varphi(\widehat{G}) - \varphi(G))$ as a sum of matrices, where each matrix is of the form $E^t SQ$ such that $\|Q\|_2 = O(1)$. This way, we have $\|E^t SQ\|_{2\to\infty} \leq \|E^t S\|_{2\to\infty} \cdot O(1)$, and $\|E^t S\|_{2\to\infty}$ is bounded by a lemma from [MZ22b].

Let $D \stackrel{\text{def}}{=} \psi(\widehat{G}) - \psi(G) = A(EG + GE + E^2) + BE$ and write $F \stackrel{\text{def}}{=} \psi(G), \widehat{F} \stackrel{\text{def}}{=} \psi(\widehat{G})$. Then

$$\varphi(\widehat{G}) - \varphi(G) = \psi(\widehat{G})^r - \psi(G)^r = (F + D)^r - F^r$$
$$= \underbrace{F^{r-1}D + F^{r-2}D\widehat{F} + \cdots + FD\widehat{F}^{r-2}}_{\stackrel{\text{def}}{=}M} + D\widehat{F}^{r-1},$$

where the last step is a decomposition based on the first location of $D$ in the product terms. And

$$D\widehat{F}^{r-1} = D(D + F)^{r-1} = D^r + \underbrace{DF\widehat{F}^{r-2} + D^2 F\widehat{F}^{r-3} + \cdots + D^{r-1}F}_{\stackrel{\text{def}}{=}M'}.$$

That is, $E(\varphi(\widehat{G}) - \varphi(G)) = EM + ED^r + EM'$. We bound the three terms respectively.

Here we first list some definitions and estimations of the quantities involved.

- According to Proposition 2.3, with probability at least $1 - n^{-3}$, we have $\|E\|_2 \leq C_0\sigma\sqrt{n}$, where $C_0$ is a constant. In the following argument, we always assume this holds.

- $\mu \stackrel{\text{def}}{=} (p - q)n/k$. By our assumption on $n$, we have $\mu \geq (100 + C_0)\sqrt{np}\log^6 n$.

- $A = -\frac{1}{\lambda_1(G)\mu}, B = (\frac{1}{\lambda_1(G)} + \frac{1}{\mu}), r = \log n; \lambda_1(G) > \mu$, and thus $B \leq \frac{2}{\mu}, |A| \leq \frac{1}{\mu^2}$.

- By Claim 3.1, $\|F\|_2 \leq 1 + \frac{1}{4\log n}, \left\|\widehat{F}\right\|_2 \leq 1 + \frac{1}{4\log n}$. By Proposition 2.5, $\|F\|_2^t, \left\|\widehat{F}\right\|_2^t \leq 2, \forall t \leq \log n$.

**Upper bound of** $\|EM\|_{2\to\infty}$  Note that $\|EF^tD\|_{2\to\infty} \leq \|EF\|_{2\to\infty}\|F\|_2^{t-1}\|D\|_2$, and $\|F\|_2^{t-1} \leq 2$ for all $t \leq r$. Moreover, $\|D\|_2 \leq |A|(2\|E\|_2\|G\|_2 + \|E\|_2^2) + B\|E\|_2 \leq 3\frac{\|E\|_2}{\mu} + \frac{\|E\|_2^2}{\mu^2} + \leq 4\frac{\|E\|_2}{\mu} \leq 4(\log^6 n)^{-1} < \frac{1}{\log^3 n}$. And the following lemma gives an upper bound of $\|EF\|_{2\to\infty}$.

**Lemma 4.1.** $\mathbf{Pr}_E\left[\|EF\|_{2\to\infty} \leq 10(\sqrt{kp\log n} + \sqrt{\log n})\right] \geq 1 - 2n^{-2}$.

Therefore, by union bound, we have the following holds with probability at least $1 - n^{-1}$:

$$\|EM\|_{2\to\infty} \leq r \cdot 10(\sqrt{kp\log n} + \sqrt{\log n}) \cdot 2 \cdot \frac{1}{\log^3 n} \leq \frac{40(\sqrt{kp}+1)}{\log n}. \tag{4}$$

**Upper bound of** $\|ED^r\|_{2\to\infty}$  Since $\|D\|_2 < \frac{1}{\log^3 n}$, we have

$$\|ED^r\|_{2\to\infty} \leq \|E\|_{2\to\infty}\|D\|_2^r \leq \sqrt{n} \cdot (\log^3 n)^{-\log n} < \frac{1}{n}. \tag{5}$$

**Lemma 4.2** (Upper bound of $\|EM'\|_{2\to\infty}$). *With probability $1 - O(n^{-2})$ (over the choice of E), we have $\|EM'\|_{2\to\infty} \leq 6C_2\sqrt{kp}\log^2 n$, where $C_2 = 10^6$ is a constant.*

Finally, combining Equation (4), Equation (5), and the above lemma, we conclude that with probability at least $1 - O(n^{-2})$,

$$\left\|E(\varphi(\widehat{G}) - \varphi(G))\right\|_{2\to\infty} \leq \frac{40(\sqrt{kp}+1)}{\log n} + \frac{1}{n} + 6C_2\sqrt{kp}\log^2 n \leq 7C_2(\sqrt{kp}\log^2 n + \frac{1}{\log n}).$$

This establishes Lemma 3.3.

Proofs of Lemma 4.1 and Lemma 4.2 are deferred to Appendix D.

## Acknowledgments and Disclosure of Funding

We are grateful to anonymous NeurIPS reviewers for their helpful comments. This research is supported by NSF CAREER award 2141536.

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

# A Useful inequalities

**Proposition A.1** (Chernoff bound). *Let $X_1, \ldots, X_m$ be i.i.d random variables that can take values in $\{0, 1\}$, with $\mathbf{E}[X_i] \leq p$ for $1 \leq i \leq m$. Then it holds that*

$$\mathbf{Pr}\left[\left|\sum_{i=1}^{n} X_i - mp\right| \geq t\right] \leq \exp\left(-\frac{3t^2}{mp}\right).$$

**Proposition A.2** (Hoeffding bound). *Let $X_1, \ldots, X_m$ be independent random variables such that $a_i \leq X_1 \leq b_i$, and write $S \overset{\text{def}}{=} \sum_{i=1}^{m} X_i$. Then it holds that*

$$\mathbf{Pr}\left[|S - \mathbf{E}[S]| > t\right] \leq 2\exp\left(-\frac{2t^2}{\sum_{i=1}^{m}(b_i - a_i)^2}\right).$$

**Definition A.1.** Let $X$ be a Bernoulli random variable with parameter $p$, i.e., $\mathbf{Pr}[X = 1] = p, \mathbf{Pr}[X = 0] = 1 - p$. The random variable $Y \overset{\text{def}}{=} X - p = X - \mathbf{E}[X]$ is called *centered Bernoulli random variable with parameter $p$*.

**Proposition A.3** (Adapted from [MZ22b]). *Let $S \in \mathbb{R}^{n \times n}$, and let $E = (\xi_{ij})$ be an $n \times n$ symmetric random matrix, where*

$$\{\xi_{ij} : 1 \leq i \leq j \leq n\}$$

*are independent, centered Bernoulli random variables with parameter at most $\alpha$ for all $i, j$. Suppose that every entry of $S$ takes value in $[-\beta, \beta]$, and each column of $S$ has at most $\gamma$ non-zero entries. Then for every $t \in [\log n]$, it holds that*

$$\mathbf{Pr}\left[\left|(E^t S)_{ij}\right| > (\log n)^{5t} C_t\right] = O(n^{-4}), \forall i, j \in [n],$$

*where*

$$C_t \overset{\text{def}}{=} 500\beta\sqrt{\alpha}\sqrt{\gamma} \cdot \left(100\sqrt{n\alpha}\right)^{t-1}.$$

*By union bound,*

$$\mathbf{Pr}\left[\left\|E^t S\right\|_{2 \to \infty} > \sqrt{n}(\log n)^{5t} C_t\right] = O(n^{-2}).$$

*Remark* A.1. The parameter $\alpha$ is determined by $E$, which equals to $p$ in our case. The above bound is particularly useful when $\beta, \gamma$ are small, that is, we want the matrix $S$ to have either small entries or sparse columns.

**Proposition A.4** (Proposition 2.5 restated). *For $a \in [0, 2]$ and $r \in \mathbb{N}$, if $|a - 1| \leq \delta < \frac{1}{2r}$, then $|a^r - 1| \leq 2r\delta$.*

*Proof.* Let $x = a - 1 \in [-\delta, \delta]$. If $0 \leq a \leq 1$, we have $1 \geq a^r = (1 + x)^r \geq 1 + rx \geq 1 - r\delta$. If $1 < a < 1 + 1/r$, then $0 < x < 1/r$ and hence

$$1 \leq a^r = (1 + x)^r = \sum_{i=0}^{r} \binom{r}{i} x^i \leq \sum_{i=0}^{r} r^i x^i < \sum_{i=0}^{\infty} (rx)^i = \frac{1}{1 - rx} = 1 + \frac{rx}{1 - rx} \leq 1 + 2r\delta.$$

$\square$

# B Bounding the Deviation Term

*Proof of Lemma 3.1.* Our assumption on $n$ implies that $(p - q)n/k > 100C_0\sigma\sqrt{n}$. By Proposition 2.3, with probability at least $1 - n^{-3}$, we have

$$\|E\|_2 \leq C_0\sigma\sqrt{n} \leq 0.01(p - q)n/k.$$

According to Equation (2) and $s_u \geq \frac{n}{2k}$, we have

$$\left\|(P_{\widehat{G}} - I)G_u\right\| \leq \frac{2\|E\|_2}{\sqrt{s_u}} \leq 0.04(p - q)n/k.$$

$\square$

## C  Polynomial Approximation

The proof of Theorem 3.1 rely on the following result on rank-one pertuebation.

**Proposition C.1** (Eigenvalues under rank-one perturbation, Theorem 1 in [BNS79])**.** *Let $C = D + \rho z z^T$, where $D$ is diagonal, $\|z\|_2 = 1$. Let $d_1 \geq d_2 \geq \cdots \geq d_n$ be the eigenvalues of $D$, and let $\widetilde{d}_1 \geq \widetilde{d}_2 \geq \cdots \geq \widetilde{d}_n$ be the eigenvalues of $C$. Then*

$$\widetilde{d}_i = d_i + \rho \mu_i, \quad 1 \leq i \leq n,$$

*where $\sum_{i=1}^{n} \mu_i = 1$ and $0 \leq \mu_i \leq 1$.*

*Proof of Theorem 3.1.* Let $\chi_i \in \{0,1\}^V$ be the indicator vector for $V_i$, i.e., $\chi_i(u) = 1$ iff $u \in V_i$. It is easy to see that the eigenvectors of $H$ are $\frac{1}{\sqrt{s_1}}\chi_1, \ldots, \frac{1}{\sqrt{s_k}}\chi_k$. Write $U = \left( \frac{1}{\sqrt{s_1}}\chi_1, \ldots, \frac{1}{\sqrt{s_k}}\chi_k \right) \in \mathbb{R}^{V \times V}$, $D = \mathrm{diag}((p-q)s_1, \ldots, (p-q)s_k, 0, \ldots, 0)$, then we have $H = UDU^\top$. Note that $\mathbf{1}_n = U(\sqrt{s_1}, \ldots, \sqrt{s_n})^\top$, and hence

$$G = H + q\mathbf{1}_n \mathbf{1}_n^\top = U(D + \rho z z^\top)U^\top,$$

where $\rho = nq, z = \frac{1}{\sqrt{n}}(\sqrt{s_1}, \ldots, \sqrt{s_n})^\top$. This means the eigenvalues of $G$ are the same as those of $D + \rho z z^\top$. Since $\|z\| = 1$, Item 1 follow directly from Proposition C.1. To see Item 2, we use the Rayleigh quotient characterization of the largest eigenvalue:

$$\lambda_1(G) = \max_v \frac{v^\top G v}{\|v\|^2} \geq \frac{\mathbf{1}_n^\top G \mathbf{1}_n}{n} = \frac{\sum_{u,v \in X} G_{uv}}{n} = \frac{n^2 q + (p-q) \cdot (s_1^2 + \cdots + s_k^2)}{n}$$

$$\geq nq + (p-q)\frac{n}{k}.$$

where the last inequality follows from $\frac{n}{k} = \frac{1}{k}\sum_{i=1}^{k} s_i \leq \sqrt{\sum_{i=1}^{k} s_i^2/k}$. $\qquad\square$

*Proof of Claim 3.1.* The assumption on $n$ in Lemma 3.2 implies that $\mu = (p-q)n/k \geq 100 C_0 \sigma \sqrt{n} \cdot \log n$. By Weyl's inequality, we have

$$\left| \widehat{\lambda}_i - \lambda_i \right| \leq \|E\|_2 \leq C_0 \sigma \sqrt{n} \leq \frac{\mu}{100 \log n}, \forall i \in [n].$$

Meanwhile, by Theorem 3.1,

$$|\lambda_i - \mu| \leq |\lambda_i(G) - (p-q)s_i| + |(p-q)s_i - \mu| \leq \frac{(p-q)n/k}{16 \log n} + \frac{(p-q)n/k}{16 \log n} \leq \frac{\mu}{8 \log n}.$$

for $i = 2, 3, \ldots, k$. Hence, write $\varepsilon \overset{\text{def}}{=} \frac{\mu}{6 \log n}$, we have

1. $\left| \widehat{\lambda}_1 - \lambda_1 \right| \leq \varepsilon$;

2. $\lambda_2, \ldots, \lambda_k, \widehat{\lambda}_2, \ldots, \widehat{\lambda}_k \in [\mu - \varepsilon, \mu + \varepsilon]$;

3. for every $i \geq k+1$, $\left| \widehat{\lambda}_i \right| \leq \varepsilon$.

First, $\psi(\lambda_1) = 1$ according to the definition of $\psi$, and hence $\varphi(\lambda_1) = 1$. As for $\widehat{\lambda}_1$,

$$\left| \psi(\widehat{\lambda}_1) - 1 \right| = \left| \psi(\widehat{\lambda}_i) - \psi(\lambda_1) \right| \leq |A|\varepsilon^2 + |2A\lambda_1 + B|\varepsilon \qquad \text{(by definition of } \psi)$$

$$\leq \frac{\varepsilon^2}{\lambda_1 \mu} + \frac{\varepsilon}{\mu} \qquad \text{(since } 2A\lambda_1 + B = \frac{1}{\lambda_1} - \frac{1}{\mu} \geq -\frac{1}{\mu})$$

$$\leq \frac{1}{36 \log^2 n} + \frac{1}{6 \log n} \qquad \text{(as } \frac{\varepsilon}{\mu} = \frac{1}{6 \log n})$$

$$< \frac{1}{4 \log n}.$$

Consequently, $|\varphi(\widehat{\lambda}_1) - 1| < \frac{2r}{4\log n} \leq 1/2$ by Proposition 2.5.

Next, for $a \in \left\{\lambda_2, \ldots, \lambda_k, \widehat{\lambda}_2, \ldots, \widehat{\lambda}_k\right\}$, the argument is similar:

$$|\psi(a) - 1| = |\psi(a) - \psi(\mu)| \leq |A|\varepsilon^2 + |2A\mu + B|\varepsilon \leq \frac{\varepsilon^2}{\lambda_1\mu} + \frac{\varepsilon}{\mu} < \frac{1}{4\log n},$$

where the second inequality follows from $2A\mu + B = \frac{1}{\mu} - \frac{1}{\lambda_1} \leq \frac{1}{\mu}$. This yields $|\varphi(a) - 1| \leq 1/2$ by Proposition 2.5.

Finally, for $i \geq k + 1$, it holds that

$$\left|\psi(\widehat{\lambda}_i)\right| \leq |A|\varepsilon^2 + B\varepsilon = \frac{\varepsilon^2}{\lambda_1\mu} + \frac{\varepsilon}{\mu} < \frac{1}{4\log n},$$

which means $\left|\varphi(\widehat{\lambda}_i)\right|^r = \left|\psi(\widehat{\lambda}_i)\right|^r < \left(\frac{1}{4\log n}\right)^{\log n} < n^{-\log\log n}$. $\qquad\square$

## D   Bounding the Noise Term

*Proof of Lemma 4.1.* The lemma readily follows from the following entrywise bound and Chernoff bound.

**Claim D.1** (Entries of $\psi(G)$). *For every $u, v \in X$, if $u, v \in V_\ell$ for some $\ell$, then $0 \leq F_{uv} \leq \frac{5k}{n}$; otherwise, $|F_{uv}| \leq \frac{10}{n}$.*

We decompose $F = F' + F''$, where $F'$ is the intra-cluster part, i.e., $F'_{uv} = F_{uv}$ if $u, v \in V_\ell$ for some $\ell$, and $F'_{uv} = 0$ otherwise. Since for every column of $F'_v$, its non-zero entries are identical and at most $5k/n$ by the above claim. Hence, every entry of $EF'$ equals to the sum of at most $2n/k$ independent, centered Bernoulli variables with parameter $p$, scaled by some factor at most $\frac{5k}{n}$. By Chernoff bound, $\mathbf{Pr}_E\left[|(EF')_{uv}| > 10\sqrt{kp\log n/n}\right] \leq n^{-4}, \forall u, v \in V$, and we have $\mathbf{Pr}_E\left[\|EF'\|_{2\to\infty} \leq 10\sqrt{kp\log n}\right] \geq 1 - n^{-2}$ by union bound. Analogously, by Hoeffding bound, $\mathbf{Pr}_E\left[\|EF''\|_{2\to\infty} \leq 10\sqrt{\log n}\right] \geq 1 - n^{-2}$. Since $\|EF\|_{2\to\infty} \leq \|EF'\|_{2\to\infty} + \|EF''\|_{2\to\infty}$, the lemma follows from the above two inequalities and union bound. $\qquad\square$

*Proof of Claim D.1.* Write $\lambda = \lambda_1(G)$ and recall that (i) $(p-q)s_u \leq 2\mu$ for all $u$ (ii) $nq + \mu \leq \lambda \leq nq + (p-q)s_1 < nq + 2\mu$, (iii) $\lambda > p \cdot \mu$. Assume that $u, v \in V_\ell$ for some $\ell$. Then

$$\begin{aligned}
F_{uv} = AG_u^\top G_v + BG_{uv} &= -\frac{nq^2 + (p^2 - q^2)s_u}{\lambda\mu} + \left(\frac{1}{\lambda} + \frac{1}{\mu}\right)p \\
&= \frac{-nq^2 - (p^2 - q^2)s_u + (p-q)(\lambda + \mu) + q(\lambda + \mu)}{\lambda\mu} \\
&= \frac{q(\mu + \lambda - nq) + (p-q)(\lambda + \mu - (p+q)s_u)}{\lambda\mu}.
\end{aligned}$$

Since $\lambda - nq \geq \mu$, the numerator is at least

$$2q\mu + (p-q)(\lambda + \mu - (p+q)s_u) = (p-q)\left(2qn/k + \lambda + \mu - (p+q)s_u\right).$$

Because $s_u \leq 2n/k, \lambda \geq nq + (p-q)n/k$, we have

$$2qn/k + \lambda + \mu - (p+q)s_u > 2qn/k + nq + (p-q)2n/k - (p+q)2n/k = (n - 2n/k)q \geq 0,$$

which means $F_{uv} \geq 0$. Meawhile,

$$F_{uv} \leq \frac{q(\mu + \lambda - nq)}{\lambda\mu} + \frac{(p-q)(\lambda + \mu)}{\lambda\mu},$$

where the first term is at most $\frac{3q}{\lambda} \leq \frac{3}{n}$ by (ii); second term is at most $\frac{2(p-q)}{\mu} \leq \frac{2k}{n}$. Therefore, $|F_{uv}| \leq \frac{5k}{n}$.

For the second part, assume that $u, v$ are not in the same cluster. Then

$$F_{uv} = AG_u^\top G_v + BG_{uv} = -\frac{nq^2 + (pq - q^2)(s_u + s_v)}{\lambda\mu} + \left(\frac{1}{\lambda} + \frac{1}{\mu}\right)q.$$

Hence,

$$|F_{uv}| \leq \left|\frac{q(\lambda + \mu - nq)}{\lambda\mu}\right| + \left|\frac{q(p - q)(s_u + s_v)}{\lambda\mu}\right|.$$

By (ii), the first term is at most $\frac{3q}{\lambda} < \frac{3}{n}$; by (i), the second term is at most $\frac{4q}{\lambda} \leq \frac{4}{n}$; hence, $|F_{uv}| \leq \frac{10}{n}$. $\qquad\square$

**Upper bound of** $\|EM'\|_{2\to\infty}$ **(Proof of Lemma 4.2)** Write $L \overset{\text{def}}{=} A(EG + GE), R \overset{\text{def}}{=} AE^2 + BE$. Then $D^t F = (L + R)^t F = R^t F + R^{t-1}LF + R^{t-2}LDF + \cdots + RLD^{t-2} + LD^{t-1}F$. It suffices to derive a good upper bound of $\|E^\eta L\|_{2\to\infty}$ and $\|E^\eta F\|_{2\to\infty}$, as $R^w$ can be further expressed as sum of powers of $E$. This is done by the following lemma:

**Lemma D.1.** *The following holds with probability $1 - O(n^{-2})$ over the choice of $E$: for all $\eta \leq \log n$, it holds that*

- $\|E^\eta L\|_{2\to\infty} \leq C_2\sqrt{kp}(100\sqrt{np})^{\eta-1}\log^{5\eta} n,$

- $\|E^\eta F\|_{2\to\infty} \leq C_2\sqrt{kp}(100\sqrt{np})^{\eta-1}\log^{5\eta} n,$

*where $C_2 \overset{\text{def}}{=} 10^6$ is an absolute constant.*

Specifically,

$$ED^t F = ER^t F + \sum_{i=0}^{t-1} ER^i LD^{t-1-i}F$$

$$= \underbrace{E\sum_{j=0}^{t}\binom{t}{j}A^j B^{t-j}E^{j+t}F}_{\overset{\text{def}}{=} M_t} + \underbrace{\sum_{i=0}^{t-1}E\sum_{j=0}^{i}\binom{i}{j}A^j B^{i-j}E^{i+j}LD^{t-1-i}F}_{\overset{\text{def}}{=} N_t}.$$

Note that $|A^j B^{w-j}| \leq 2^w \cdot \mu^{-(w+j)} \leq 2^w \cdot (100\sqrt{np}\log^6 n)^{-(w+j)}$. It follows that for every $t \in [r-1]$,

$$\|M_t\|_{2\to\infty} \leq \sum_{j=0}^{t}\binom{t}{j}|A^j B^{t-j}|\,\|E^{t+j+1}F\|_{2\to\infty}$$

$$\leq \sum_{j=0}^{t}\binom{t}{j}2^t \cdot (100\sqrt{np}\log^6 n)^{-(t+j)} \cdot C_2\sqrt{kp} \cdot (100\sqrt{np})^{t+j}\log^{5(t+j)} n$$

$$\leq C_2\sqrt{kp} \cdot 2^t \sum_{j=0}^{t}\binom{t}{j}(\log n)^{-(t+j)}$$

$$= C_2\sqrt{kp} \cdot 2^t(\log n)^{-t} \cdot (1 + \frac{1}{\log n})^t \leq C_2\sqrt{kp},$$

where the second inequality is by Lemma D.1, and the last step follows from Proposition 2.5. Similarly, for every $t \in [r-1]$,

$$\|N_t\|_{2\to\infty} \leq \sum_{i=0}^{t-1}\sum_{j=0}^{i}\binom{i}{j}|A^j B^{i-j}|\,\|E^{i+j+1}L\|_{2\to\infty}\,\|D^{t-1-i}F\|_2$$

$$\leq 2\sum_{i=0}^{t-1}\sum_{j=0}^{i}\binom{i}{j}|A^j B^{i-j}|\,\|E^{i+j+1}L\|_{2\to\infty}$$

$$\leq 2\sum_{i=0}^{t-1}\sum_{j=0}^{i}\binom{i}{j}2^i\cdot(100\sqrt{np}\log^6 n)^{-(i+j)}\cdot C_2\sqrt{kp}\cdot(100\sqrt{np})^{i+j}\log^{5(i+j)}n$$

$$\leq 2C_2\sqrt{kp}\sum_{i=0}^{t-1}2^i\sum_{j=0}^{i}\binom{t}{j}(\log n)^{-(i+j)}$$

$$\leq 2C_2\sqrt{kp}\cdot t \leq 2C_2\sqrt{kp}\cdot\log n,$$

where the second inequality follows from $\|D^{t-1-i}F\|_2 \leq 2$, and the third inequality is by Lemma D.1. In sum,

$$\|EM'\|_{2\to\infty} \leq \sum_{t=1}^{r-1}(\|M_t\|_{2\to\infty}+\|N_t\|_{2\to\infty})\left\|\widehat{F}\right\|_2^{r-1-t} \leq 6C_2\sqrt{kp}\log^2 n, \tag{6}$$

where in the last inequality we also use $\left\|\widehat{F}\right\|_2^t \leq 2$.

It remains to prove Lemma D.1; the proof draws on the entrywise bound in Proposition A.3.

*Proof of Lemma D.1.* Fix an $\eta \leq \log n$. Write $s^* \stackrel{\text{def}}{=} n/k$ for the ease of notation. Observe that $E^\eta L = A(E^{\eta+1}G + E^\eta GE) = A(E^{\eta+1}H + E^\eta HE) + Aq(E^{\eta+1}J_n + E^\eta J_n E)$. We can apply Proposition A.3 to $E^{\eta+1}H$ and $E^\eta H$, with $\alpha = p, \beta = p - q, \gamma = 2s^*$. That is, with probability at least $1 - O(n^{-2})$,

$$\left\|E^j H\right\|_{2\to\infty} \leq 500(\log n)^{5j}\sqrt{n}(p-q)\sqrt{p}\sqrt{2s^*}\cdot(100\sqrt{np})^{j-1}\ \text{for } j = \eta, \eta+1.$$

Our assumption on $n$ yields $\mu \geq C\sqrt{np}\log^6 n$; moreover, $|A|(p-q) \leq \frac{p-q}{\mu^2} \leq 1/s^*\cdot\frac{1}{\mu} \leq 1/s^*\cdot(C\sqrt{np}\log^6 n)^{-1}$. Therefore,

$$\left\|A(E^{\eta+1}H + E^\eta HE)\right\|_{2\to\infty}$$
$$\leq A\left(\left\|E^{\eta+1}H\right\|_{2\to\infty} + \left\|E^\eta H\right\|_{2\to\infty}\|E\|_2\right)$$
$$\leq \frac{1}{s^*}\cdot(C\sqrt{np}\log^6 n)^{-1}\cdot 500(\log n)^{5\eta+5}\cdot\sqrt{n}\cdot\sqrt{p}\cdot\sqrt{2s^*}\left((100\sqrt{np})^\eta + (100\sqrt{np})^{\eta-1}C_0\sigma\sqrt{n}\right)$$
$$\leq 500000\sqrt{np/s^*}(\log n)^{5\eta}(100\sqrt{np})^{\eta-1}. \tag{7}$$

Similarly, by applying Proposition A.3 to $E^j J_n$ with $\alpha = p, \beta = 1, \gamma = n$, we have, with probability at least $1 - O(n^{-2})$,

$$\left\|E^j J_n\right\|_{2\to\infty} \leq 500(\log n)^{5j}\cdot\sqrt{p}\cdot n\cdot(100\sqrt{np})^{j-1}, j = \eta, \eta+1.$$

Since $|Aq| = \frac{q}{\lambda}\cdot\frac{1}{\mu} \leq \frac{1}{n}\cdot(C\sqrt{n}\log^6 n)^{-1}$, we have

$$\left\|Aq(E^{\eta+1}J_n + E^\eta J_n E)\right\|_{2\to\infty}$$
$$\leq |Aq|\left(\left\|E^{\eta+1}J_n\right\|_{2\to\infty} + \left\|E^\eta J_n\right\|_{2\to\infty}\|E\|_2\right)$$
$$\leq \frac{1}{n}\cdot(C\sqrt{np}\log^6 n)^{-1}\cdot 500\cdot(\log n)^{5\eta+5}\cdot n\cdot\left((100\sqrt{np})^\eta + (100\sqrt{np})^{\eta-1}C_0\sigma\sqrt{n}\right)$$
$$\leq 500000\sqrt{p}(\log n)^{5\eta}(100\sqrt{np})^{\eta-1}. \tag{8}$$

Combining Equation (7) and Equation (8), we have, with probability at least $1 - O(n^{-2})$, $\|E^\eta L\|_{2\to\infty} \le C_2\sqrt{np/s^*}(\log n)^{5\eta}(100\sqrt{np})^{\eta-1}$ where $C_2 \overset{\text{def}}{=} 10^6$.

For the second part, we decompose $F = F' + F''$, where $F'$ is the intra-cluster part, i.e., $F'_{uv} = F_{uv}$ if $u, v \in V_\ell$ for some $\ell$; and $F'_{uv} = 0$ otherwise. Equipped with Claim D.1, we can apply Proposition A.3 on $E^\eta F'$ (with $\alpha = p, \beta = 10/s^*, \gamma = 2s^*$), and $E^\eta F''$ (with $\alpha = p, \beta = \frac{5}{n}, \gamma = n$):

$$\begin{aligned}
\|E^\eta F\|_{2\to\infty} &\le \|E^\eta F'\|_{2\to\infty} + \|E^\eta F''\|_{2\to\infty} \\
&\le 20000\log^{5\eta}\sqrt{n}\sqrt{p}\left(\frac{10}{s^*}\cdot\sqrt{2s^*} + \frac{10}{n}\cdot\sqrt{n}\right)(100\sqrt{n})^{\eta-1} \\
&\le C_2(\log n)^{5\eta}\sqrt{np/s^*}(100\sqrt{n})^{\eta-1},
\end{aligned}$$

where the second inequality holds with probability at least $1 - O(n^{-2})$. The lemma follows from a union bound over all $\eta \le \log n$. $\qquad\square$

