# On the Power of SVD in Clustering Problems

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

&\le 2C_2\sqrt{kp}\cdot t\le 2C_2\sqrt{kp}\cdot\log n,
\end{aligned}
$$

where the second inequality follows from $\left\|D^{t-1-i}F\right\|_2 \le 2$, and the third inequality is by Lemma D.1. In sum,

$$
\|EM'\|_{2\to\infty}\le\sum_{t=1}^{r-1}(\|M_t\|_{2\to\infty}+\|N_t\|_{2\to\infty})\left\|\widehat{F}\right\|_2^{r-1-t}\le 6C_2\sqrt{kp}\log^2 n, \tag{6}
$$

where in the last inequality we also use $\left\|\widehat{F}\right\|_2^t\le 2$.

The proof of Lemma D.1 draw on the entrywise bound in Proposition A.3.

*Proof of Lemma D.1.* Fix an $\eta\le\log n$. Write $s^*\overset{\text{def}}{=}n/k$ for the ease of notation. Observe that $E^\eta L = A(E^{\eta+1}G + E^\eta GE) = A(E^{\eta+1}H + E^\eta HE) + Aq(E^{\eta+1}J_n + E^\eta J_n E)$. We can apply Proposition A.3 to $E^{\eta+1}H$ and $E^\eta H$, with $\alpha = p, \beta = p-q, \gamma = 2s^*$. That is, with probability at least $1 - O(n^{-2})$,

$$
\left\|E^j H\right\|_{2\to\infty}\le 500(\log n)^{5j}\sqrt{n}(p-q)\sqrt{p}\sqrt{2s^*}\cdot(100\sqrt{np})^{j-1}\text{ for }j=\eta,\eta+1.
$$

Our assumption on $n$ yields $\mu\ge C\sqrt{np}\log^6 n$; moreover, $|A|(p-q)\le\frac{p-q}{\mu^2}\le 1/s^*\cdot\frac{1}{\mu}\le 1/s^*\cdot(C\sqrt{np}\log^6 n)^{-1}$. Therefore,

$$
\begin{aligned}
&\left\|A(E^{\eta+1}H + E^\eta HE)\right\|_{2\to\infty}\\
&\le A\left(\left\|E^{\eta+1}H\right\|_{2\to\infty}+\left\|E^\eta H\right\|_{2\to\infty}\|E\|_2\right)\\
&\le\frac{1}{s^*}\cdot(C\sqrt{np}\log^6 n)^{-1}\cdot 500(\log n)^{5\eta+5}\cdot\sqrt{n}\cdot\sqrt{p}\cdot\sqrt{2s^*}\left((100\sqrt{np})^\eta+(100\sqrt{np})^{\eta-1}C_0\sigma\sqrt{n}\right)\\
&\le 500000\sqrt{np/s^*}(\log n)^{5\eta}(100\sqrt{np})^{\eta-1}.
\end{aligned} \tag{7}
$$

Similarly, by applying Proposition A.3 to $E^j J_n$ with $\alpha = p, \beta = 1, \gamma = n$, we have, with probability at least $1 - O(n^{-2})$,

$$
\left\|E^j J_n\right\|_{2\to\infty}\le 500(\log n)^{5j}\cdot\sqrt{p}\cdot n\cdot(100\sqrt{np})^{j-1},\ j=\eta,\eta+1.
$$

Since $|Aq| = \frac{q}{\lambda}\cdot\frac{1}{\mu}\le\frac{1}{n}\cdot(C\sqrt{n}\log^6 n)^{-1}$, we have

$$
\begin{aligned}
&\left\|Aq(E^{\eta+1}J_n + E^\eta J_n E)\right\|_{2\to\infty}\\
&\le|Aq|\left(\left\|E^{\eta+1}J_n\right\|_{2\to\infty}+\left\|E^\eta J_n\right\|_{2\to\infty}\|E\|_2\right)\\
&\le\frac{1}{n}\cdot(C\sqrt{np}\log^6 n)^{-1}\cdot 500\cdot(\log n)^{5\eta+5}\cdot n\cdot\left((100\sqrt{np})^\eta+(100\sqrt{np})^{\eta-1}C_0\sigma\sqrt{n}\right)\\
&\le 500000\sqrt{p}(\log n)^{5\eta}(100\sqrt{np})^{\eta-1}.
\end{aligned} \tag{8}
$$

Combining Equation (7) and Equation (8), we have, with probability at least $1 - O(n^{-2})$,

$\|E^\eta L\|_{2\to\infty} \le C_2 \sqrt{np/s^*}(\log n)^{5\eta}(100\sqrt{np})^{\eta-1}$ where $C_2 \stackrel{\text{def}}{=} 10^6$.

For the second part, we decompose $F = F' + F''$, where $F'$ is the intra-cluster part, i.e., $F'_{uv} = F_{uv}$ if $u, v \in V_\ell$ for some $\ell$; and $F'_{uv} = 0$ otherwise. Equipped with Claim D.1, we can apply Proposition A.3 on $E^\eta F'$ (with $\alpha = p, \beta = 10/s^*, \gamma = 2s^*$), and $E^\eta F''$ (with $\alpha = p, \beta = \frac{5}{n}, \gamma = n$):

$$\begin{aligned}
\|E^\eta F\|_{2\to\infty} &\le \|E^\eta F'\|_{2\to\infty} + \|E^\eta F''\|_{2\to\infty} \\
&\le 20000 \log^{5\eta} \sqrt{n}\sqrt{p}\left(\frac{10}{s^*} \cdot \sqrt{2s^*} + \frac{10}{n} \cdot \sqrt{n}\right)(100\sqrt{n})^{\eta-1} \\
&\le C_2(\log n)^{5\eta}\sqrt{np/s^*}(100\sqrt{n})^{\eta-1},
\end{aligned}$$

where the second inequality holds with probability at least $1 - O(n^{-2})$. The lemma follows from a union bound over all $\eta \le \log n$. $\qquad\square$