# OpenReview forum: "On the Power of SVD in the Stochastic Block Model"
_NeurIPS.cc/2023/Conference — NeurIPS 2023 poster_

### Official Review · Reviewer_b4S2 · 2023-06-21

**Soundness:** 3 good
**Presentation:** 2 fair
**Contribution:** 3 good
**Rating:** 7
**Confidence:** 3

**Summary:**

This paper studies the power of vanilla-SVD algorithm, algorithm without any pre-processing or post-trimming steps, in the symmetric stochastic block model and proves it recovers all clusters in the balanced case, which answers an open question in [Vu18].

**Strengths:**

The main contribution of this work lies in its demonstration of the effectiveness of a truly vanilla SVD algorithm in recovering all clusters of the symmetric stochastic block model in the balanced case with high probability theoretically. Some noticeable distinctions from existing works include a truly vanilla SVD algorithm without any additional pre-processing steps and the ability of handling of cluster numbers $k = \omega(1)$. Also, from a technical standpoint, the authors introduce a novel "polynomial approximation + entrywise analysis" approach, which simplifies the analysis of eigenspace perturbation by reducing it to the analysis of a simple polynomial under perturbation thus makes the analysis more robust and requires less structure.


**Weaknesses:**

The presentation of this work could be improved. First, it might be better to explicitly mention in the Abstract and Section 1that the analysis is within the balanced case and not directly extended to other cases. Second, [MZ22b] also claims answering the open question in [Vu18] in the balanced case. The authors point out that the centered-SVD algorithm proposed in [MZ22b] is not truly vanilla, containing a pre-processing centering step which depends on the knowledge of $q$. But it would still be beneficial to compare the condition and bounds in [MZ22b] to have a more complete understanding. Third, adding more background information and necessary theoretical knowledge would make this work more accessible to a broader audience. Lastly, the analysis in Section 4 may be better positioned prior to Section 3.3, which already wraps up the entire analysis, or included within the supplementary material.

Others:
- In Lemma 3.1, it should be $1 - n^{-3}$ rather than $1- n^3$.

**Questions:**

This work argues that its analysis can handle the case of $k = \omega(1)$ while [MZ22b] considers a stronger case of $k = \omega(\log n)$. Can the analysis in this work also handle the case of $k = \omega(\log n)$?

**Limitations:**

There seems to be no discussion of limitations or potential societal impact.

---

> ### Author Rebuttal · Authors · 2023-08-04
>
> We thank this reviewer for the suggestions on improving our presentation. We will improve our writing accordingly.
>
> **Regarding the question about  $k=\omega(\log n)$:**
>
> Yes, our proof works perfectly in this case. As long as the parameters $n,p,q,k$ satisfy the requirement in Theorem 1,  our algorithm is able to recover the clusters. For example, if $p=0.51, q=0.49, k = n^{0.1}$, our algorithm can recover all clusters.

---

> > ### Comment · Reviewer_b4S2 · 2023-08-14
> >
> > Thanks for the response. I maintain my acceptance recommendation.

---

> > > ### Author Response · Authors · 2023-08-14
> > >
> > > Thank you for reviewing our response and maintaining your acceptance recommendation. We appreciate your evaluation and support :)

---

### Official Review · Reviewer_Qju9 · 2023-06-26

**Soundness:** 3 good
**Presentation:** 2 fair
**Contribution:** 2 fair
**Rating:** 5
**Confidence:** 2

**Summary:**

In order to understand the behavior of spectral steps in clustering problems, this paper studies the power of vanilla-SVD algorithm in the SBM. This work shows that vanilla-SVD algorithm recovers all clusters correctly in the symmetric setting.

**Strengths:**

1. To theoretically understand the power of practically successful vanilla spectral algorithms, this paper studies SBM as a preliminary demonstration. Then, some proofs are presented for the proposed Theorem 1.1.
2. This work shows that vanilla algorithms is indeed a clustering algorithm in SSBM for a wide range of parameters.
3. The authors give another analysis on matrix perturbation with random noise.
4. Detailed comparisons with existing analysis for vanilla spectral algorithms in SBM are presented in this paper.

**Weaknesses:**

1. In real applications, it is common in practice that spectral embeddings obtained by spectral clustering algorithm need a post-processing, e.g., k-means. It is reasonable that spectral embeddings themselves are clustering-friendly. Thus, the contributions of analyzing vanilla spectral methods could be highlighted.

2. From Section 1.3, it can be observed that there is limited work on the vanilla spectral algorithm. Could the authors provide an explanation for the reasons behind analyzing the parameters in the SSBM setting in this paper?

3. From the comparisons, it is apparent that there are some issues with the related theoretical analyses, such as [AFWZ20], [EBW18], [PPV+19], etc. Since these weaknesses are evident, could the authors consider providing quantitative experimental analysis in addition to the theoretical analysis?

4. Between lines 48 and 49, '...can be view as a fixed matrix...' should be corrected to '...viewed...'.

5. The coherence in the review content in the Introduction section is insufficient. The authors seem to focus more on the mathematical advancements rather than considering the theoretical and practical applications of spectral methods in the field of machine learning, especially the related work in both theoretical understandings and practical applications.

6. The argument is verified using a specific setting in a typical model, SSBM,  to demonstrate that vanilla spectral algorithm is powerful and prectically successful. It may not be comprehensive enough.

**Questions:**

1. Experiments are suggested to conduct from the perspectives of practical applications.
2. Considering this paper as purely theoretical work, then based on Weaknesses #3, it is recommended to provide quantitative experimental analysis to supplement and demonstrate the limitations of the theoretical analysis of the comparison methods.
3. The arguments of this paper need verifying by more settings, not only in symmetric version of SBM.
4. It is recommended to use reference citation format that complies with the guidelines specified by this conference. Additionally, it would be beneficial to gain a better understanding of the contributions of spectral methods in the field of machine learning or data mining.

**Limitations:**

The authors have adequately addressed the limitations.

---

> ### Author Rebuttal · Authors · 2023-08-04
>
> Thanks for pointing out the typo and your suggestion on the citation format; we shall certainly improve our paper accordingly.
>
> **Regarding experiments (Weakness #3 and Question #1, #2):**
>
> We study the same algorithm (SVD) as in  [AFWZ20], [EBW18] [PPV+19] and we give better analysis. Since it is the same algorithm, we are not sure how to run quantitative experimental analysis for comparison. Please let us know if you have any suggestion. For practical applications, the fact that SVD is widely-used in machine learning and data mining has already validated the effectiveness of the algorithm, so we only focus on theoretical analysis.
>
> **Regarding other issues:**
>
> - **Weakness #1.**  Even if post-processing, say K-means, is used in practice. There are a lot of practical observations that, in some scenarios, vanilla spectral algorithm + K-means is much better than K-means itself.  (There are many references. If you want, we can provide some of them.) The main reason is that vanilla spectral algorithms can remove noise in these applications. This phenomenon is exactly what we have analyzed in the SSBM model. We think this is also a good contribution.
>
> - **Weakness #2 #6 & Question #3.** SBM is a well-known benchmark for clustering tasks. SSBM is indeed a further simplification, but it is a good starting point. Analyzing parameter settings in SSBM could help understand under what conditions SVD works well theoretically.  As you suggested in Weakness #6 and Question #3, we expect future works to analyze more general settings. But even for SSBM, it is highly non-trivial. This problem has been open for several years, and some great mathematicians, such as the authors of [AFWZ20], [EBW18],[Vu18], [PPV+19]  didn’t solve it.
>
> - **Weakness #5.** We will definitely improve our introduction part to highlight the relevance to machine learning and append more related works.

---

> > ### Comment · Reviewer_Qju9 · 2023-08-14
> >
> > Thanks for the authors' reply. I have increased the score.

---

> > > ### Author Response · Authors · 2023-08-14
> > >
> > > We sincerely appreciate your willingness to reconsider based on our rebuttal. Your feedback has been invaluable to us.

---

### Official Review · Reviewer_bCxf · 2023-07-05

**Soundness:** 3 good
**Presentation:** 2 fair
**Contribution:** 3 good
**Rating:** 7
**Confidence:** 2

**Summary:**

This paper investigates the effectiveness of the vanilla-SVD algorithm in the stochastic block model (SBM) and demonstrates that it can accurately recover all clusters in the symmetric setting. The authors address an open question raised by Van Vu in the symmetric setting.


**Strengths:**

1. The vanilla algorithms employed in this study are applicable to a wide range of parameters, surpassing previous limitations that only allowed analysis on a constant number of clusters.

2. They also provide a novel analysis on  matrix perturbation with random noise.


**Weaknesses:**

1. The paper lacks experimental evaluation to demonstrate the effectiveness of the proposed scheme.

2. The time and space complexity analysis are not compared with previous schemes, which could provide insights into the efficiency of the proposed approach.

3. Lots of proofs are omitted in the main paper (I understand that this is due to the space limit). Maybe the theory conferences like COLT is a better venue for this paper?

**Questions:**

1. What advantages does the SVD approach in this paper offer compared to the terminal version of the Johnson-Lindenstrauss Lemma mentioned in ``Optimal terminal dimensionality reduction in Euclidean space''?

2. Is the proposed  scheme  efficient for large $n$ and $k$? If $n$ and $k$ are large, can the algorithm handle them efficiently?


**Limitations:**

Overall, this paper provides some valuable contributions to the study of the vanilla-SVD algorithm in the SBM. However, addressing the weaknesses mentioned above, such as conducting experiments, comparing complexities, and improving readability, would enhance the quality and impact of the research.

---

> ### Author Rebuttal · Authors · 2023-08-04
>
> We thank this reviewer for useful comments and questions.
>
> **Regarding weakness:**
>
> We did not propose a new scheme. Instead, we provided a theoretical analysis for a very popular algorithm — SVD (singular value decomposition). Given this algorithm has already been widely used in practice. We did not run experiments further to demonstrate the effectiveness.
>
> The time and space complexity of our algorithm is simply the same as the time and space complexity of SVD.
>
> All proofs are included in the supplementary. Even though the main paper has a page limitation, we will prepare an improved full version in arXiv once this paper has been accepted. COLT is a good conference, but we also love NeurIPS :-)
>
> **Regarding questions:**
>
> **Comparison to JL projections.** JL is also a good dimensionality reduction algorithm. However, it only preserves $\ell_2$ norm distance after projection. By contrast, SVD can do something more, namely, SVD also filters the noise. In fact, this is the main reason why SVD can recover clusters in SBM, but JL projection can not.
>
> **Efficiency when $k$ and $n$ are large.** The main part of our algorithm is to find the top $k$ eigenvectors of a $n\times n$ matrix. This can be implemented within $O(k n^2)$ time and $O(n^2)$ space.

---

> > ### Comment · Reviewer_bCxf · 2023-08-14
> >
> > Thanks for the response. The rebuttal addresses some of my concerns. I am willing to increase my grade slightly.

---

> > > ### Author Response · Authors · 2023-08-14
> > >
> > > Thank you for considering our rebuttal and re-evaluating our submission. We genuinely appreciate your feedback!

---

### Official Review · Reviewer_89Uu · 2023-07-06

**Soundness:** 3 good
**Presentation:** 3 good
**Contribution:** 3 good
**Rating:** 5
**Confidence:** 4

**Summary:**

The manuscript mention that the paper contributes by providing a theoretical understanding of the power of vanilla spectral algorithms in clustering problems, specifically in the stochastic block model (SBM).  It also presents a novel analysis of matrix perturbation with random noise.  These contributions suggest that the document offers new insights into the application of SVD in clustering problems, particularly in the context of SBM. It focuses on the stochastic block model (SBM), a benchmark for clustering, and treats it as a form of vector clustering. The manuscript proposes and analyzes a vanilla-SVD algorithm for graph clustering and demonstrates its effectiveness in SBM.

**Strengths:**

The main topic of the manuscript is the analysis of the power of Singular Value Decomposition (SVD) in clustering problems. The manuscript discusses the use of dimensionality reduction techniques, specifically PCA and SVD, to improve clustering results in high-dimensional datasets. It explains that classical clustering algorithms like K-means may perform poorly in such datasets due to the curse of dimensionality. Spectral methods like PCA and SVD have been observed to significantly enhance clustering results. The manuscript explores the reasons behind this improvement, including filtering noise from high-dimensional data. It focuses on the stochastic block model (SBM), a benchmark for clustering, and treats it as a form of vector clustering. The manuscript proposes and analyzes a vanilla-SVD algorithm for graph clustering and demonstrates its effectiveness in SBM. The authors present their results, including a clustering algorithm and an analysis of matrix perturbation with random noise. They compare their approach with existing analysis methods and highlight the advantages of their approach. The document concludes with a proof outline and technical contributions, outlining the steps taken to analyze the power of vanilla spectral algorithms for clustering.



**Weaknesses:**

There are many formulas that make the readability not very strong. Some experiments should be designed to evaluate the significance of the model.

**Questions:**

1] The paper focuses on the power of the vanilla-SVD algorithm in the stochastic block model (SBM).  It would be interesting to know if the algorithm's effectiveness is limited to the SBM or if it can be applied to other clustering problems as well.  Is there any generalization of its performance to different datasets?
2] The authors mention that the lack of theoretical analysis for vanilla spectral algorithms is due to technical obstacles.  Could you elaborate on these obstacles and explain why the theoretical analysis is challenging?  What are the main difficulties in analyzing vanilla spectral algorithms?
3] The authors state that the lack of theoretical analysis is partly due to the simplicity of vanilla spectral algorithms, which are not specifically designed for theoretical models like SBM.  How do these algorithms compare to more complicated algorithms designed for SBM in terms of performance and practicality?  Are there any trade-offs between simplicity and performance?
4] The authors mention the potential application of vanilla spectral algorithms in real-world data, but there is no analysis or discussion on this topic.  Can you provide any insights into the behavior of these algorithms on real data?  How do they perform in different applications?  Are there any challenges or limitations when applying vanilla spectral algorithms in practice?


**Limitations:**

This work should design some experiments to fully illustrate the application value of the model.

---

> ### Author Rebuttal · Authors · 2023-08-04
>
> We really appreciate this review’s feedback and questions. We answer them below.
>
> **Regarding the weakness:**
>
> Since this is a theoretical paper, some formulas are necessary. However, we will try our best to make this paper accessible to most readers. In fact, results and proofs in random matrix theory are often mathematically intensive. Compared to previous papers, our analysis is quite concise. For example, the paper we cited [AFWZ20] (https://arxiv.org/pdf/1709.09565.pdf) has 58 pages and way more formulas.
>
> The SBM model is a well-known benchmark for graph clustering problems (according to wikipedia, https://en.wikipedia.org/wiki/Stochastic_block_model). So we didn’t run experiments to validate this model.
>
>
> **Regarding the questions:**
>
> 1] Regarding the algorithm's effectiveness in other clustering problems.
>
> This paper focuses on vanilla spectral algorithms — specifically SVD (singular value decomposition) algorithm, which is already a very popular algorithm in practice.  We did not propose a new algorithm. Instead, we provided new theoretical insight into why this widely-used algorithm is successful.
>
> 2] On the obstacles of theoretical analysis for vanilla spectral algorithms.
>
> We have discussed on major reasons in our paper. One main reason is that many previous analyses used a key component called Davis-Kahan theorem. Davis-Kahan theorem is suboptimal in analyzing vanilla spectral algorithms in SBM (please see Page 4 of our paper). In fact, such obstacles have also been discussed in previous papers. The reason that we can analyze SVD is that we fully avoid Davis-Kahan theorem. We believe this is a highly non-trivial step since a majority of papers in random matrix theory used Davis-Kahan (or its variants) to analyze matrix perturbations. Our new approach has a potential to give more applications in different settings.
>
> 3] Compared to more complicated algorithms designed for SBM in terms of performance and practicality.
>
> These complicated algorithms perform well in SBM. However, they were not popular in practice because their performance on real datasets is no better (usually worse) than vanilla ones. In contrast, we analyzed vanilla spectral algorithms such as PCA and SVD which are successful in practice.
>
>
>  4] Application of vanilla spectral algorithms in real-world data.
>
> Vanilla algorithms such as PCA and SVD (the algorithm we analyzed in this paper) are widely used in practice. There are many papers/blogs which have discussed their performance in all kinds of real scenarios. Given this reason, we didn’t repeat these discussions. Instead, we focus on the performance of SVD in theoretical models.
>
> Overall, thanks again for these comments and questions :-)

---

### Official Review · Reviewer_qMce · 2023-07-27

**Soundness:** 4 excellent
**Presentation:** 4 excellent
**Contribution:** 3 good
**Rating:** 7
**Confidence:** 3

**Summary:**

This paper provides rigorous, theory-based evidence that vanilla spectral algorithms (i.e., methods that run SVD on the adjacency matrix without any further processing) succeed in finding many communities in symmetric stochastic block models. In contrast to Davis-Kahan approaches, the authors adopt an analysis similar to [MZ22b], which is inspired by power iteration. The key technical novelty of the authors' method is a new way to study eigenspace perturbation by using a polynomial approximation of the operator that projects a vector onto the space of the first k eigenvectors of the adjacency matrix.

**Strengths:**

While there are many community recovery algorithms for SBMs at this point, this paper provides an important analysis of vanilla algorithms for community recovery, showing that pre or post processing steps are unnecessary, thereby validating practical approaches. This is the key strength of the paper, in my view.

A secondary strength, which is of significance to researchers working on random matrix theory, is the use of the polynomial approximation method for the projection operator. This interestingly allows one to circumvent usage of the Davis-Kahan theorem (which is the standard way to handle eigenspace perturbation), and I expect this technique will be useful in various other settings.

The paper is also very well written, with clear comparisons to related work, and key contributions highlighted.

**Weaknesses:**

The main weakness I found was that there is little to no discussion of the condition under which Algorithm 1 recovers communities (Theorem 1.1). At face value, it seems like a similar condition as [Vu18], with a few changes, like $\sigma \sqrt{k}$ replaced by $\sqrt{kp} \log^6 n$. A few questions are:
- Is this change an artifact of the proof, or is it something more fundamental? In most situations, I'd expect the $\sqrt{\log n}$ term to dominate.
- What are commonly considered regimes for p, q, k in which this threshold is satisfied? For instance, it seems like the regime $p, q = \Theta ( \log (n) / n)$ is not covered by the theorem. What's the minimum scaling of $p, q$ that would work here? And what is the maximum number of communities that can be tolerated? Understanding such qualitative cases would make your results much more interpretable. In a similar vein, it was unclear to me whether the condition you have here is optimal in some sense, or if it's just to prove that vanilla methods succeed in just *some* regime.

Finally, I have a question related to how your work compares to [MZ22b].
- It seems that your work and [MZ22b] tackle a similar parameter regime for community recovery. How do the achievability regions compare between their work and yours? In particular, does the vanilla algorithm work in almost the same parameter regime, or is the parameter regime more restricted?

**Questions:**

- I would mention in the abstract that your main contribution is on vanilla algorithms with a large number of communities. This would emphasize the difference between your work and that of [AFWZ20].
- What should the value of $\Delta$ be in Algorithm 1?
- If there is space, could you elaborate on the connections between your polynomial approach and the power iteration method of [MZ22b]? From a quick read of [MZ22b] it seems there may be some similar ideas and it would be valuable for readers in the field to understand those connections further.

**Limitations:**

Limitations have been adequately addressed.

---

> ### Author Rebuttal · Authors · 2023-08-04
>
> We thank this reviewer for their appreciation of our work, useful suggestions and relevant questions.
>
> **Regarding weakness:**
>
> We believe the extra $\log^6 n$ factor can be improved by future works. This term stems from the concentration inequality we used as a black box. A refined analysis of this concentration inequality may remove this factor. However, as you also observed, the $\sqrt{\log n}$ factor seems more fundamental in that it comes from Prop. 2.4.
>
> We can handle the case where  $q = \Theta(\log n / n)$. As you observed, the difference between our paper and Vu’s paper is to replace $\sigma$ by $\sqrt{p} \cdot \log^6 n$.  If $p < 0.9$, $\sigma$ and $\sqrt{p}$ are equal up to constant, so our bound is essentially the same as [Vu18] except for the $\log^6 n$ factor. Here are some example setting of parameters:
> - In the regime $q = \Theta(\log n / n)$,  $p = \Omega(\frac{k \log^6 n}{\sqrt{n}})$ would be sufficient.
> - In the regime $p - q = \Theta(1)$, we can recover up to  $k = O\left(\frac{\sqrt{n}}{\log^6 n}\right)$ clusters.
>
> Compared to [MZ22b], we have an extra $\sqrt{\log n}$ factor because we use Prop. 2.4 as did in [Vu18].
>
> Thanks for your suggestion. We will add more explanations in our paper.
>
> ---
>
> **Regarding questions:**
>
> Thanks for the suggestion! We will improve our abstract accordingly.
>
> We can choose $\Delta := 0.8(p − q) n / k$ (please see Section 3.3).
>
> To make their analysis work, [MZ22b]’s power iteration requires the matrix to be nice, i.e., all large eigenvectors are very close. Therefore, in SSBM, they first ``shift’’ the adjacency matrix by considering $G':= G - q\boldsymbol{1}\boldsymbol{1}^\top$. Only the shifted matrix $G’$ has this nice structure and the original $G$ doesn’t have. This shifting step makes analysis simpler at the cost of not being vanilla.
>
> In contrast, we introduced a polynomial $\psi$ and analyzed the eigenspace of $G$ through the power iteration of $\psi(G)$, which is our main novelty.  There are two benefits of using this polynomial: (i) $\psi(G)$ has a nice structure; (ii) $\psi$ only appears in our analysis, keeping our algorithm vanilla.

---

### Decision · Program_Chairs · 2023-09-21

**Decision:**

Accept (poster)

**Comment:**

Generally reviewers felt positively about this paper. it resolves an important open theoretical question about stochastic block model (SBM) recovery with `vanilla' spectral algorithms. Given the theoretical importance of the SBM model, and the importance of spectral clustering methods in practice, this result should be impactful.

During discussion it was agreed that a lack of empirical results in the paper was not a concern, due to the paper addressing a theoretical question about a very empirically well-established approach.